# 3D reconstruction of bird flight trajectories using a single video camera

M. V. Srinivasan[ID]*[☯], H. D. Vo[☯], I. Schiffner[ID][☯]

Queensland Brain Institute, University of Queensland, St. Liucia, QLD, Australia

☯ These authors contributed equally to this work.
* m.srinivasan@uq.edu.au

**Data Availability Statement:** Link to data repository: https://figshare.com/s/ecac7955a9a9c6ed59f5.

**Funding:** (MVS): ARC Centre of Excellence in Vision Science (Grant CEO561903) https://www.arc.gov.au/grants/linkage-program/arc-centres-

## Abstract

Video cameras are finding increasing use in the study and analysis of bird flight over short ranges. However, reconstruction of flight trajectories in three dimensions typically requires the use of multiple cameras and elaborate calibration procedures. We present an alternative approach that uses a single video camera and a simple calibration procedure for the reconstruction of such trajectories. The technique combines prior knowledge of the wingspan of the bird with a camera calibration procedure that needs to be used only once in the lifetime of the system. The system delivers the exact 3D coordinates of the position of the bird at the time of every full wing extension and uses interpolated height estimates to compute the 3D positions of the bird in the video frames between successive wing extensions. The system is inexpensive, compact and portable, and can be easily deployed in the laboratory as well as the field.

## 1. Introduction

The increasing use of high-speed video cameras is offering new opportunities as well as challenges for tracking three-dimensional motions of humans and animals, and of their body parts (e.g. [1–10]).

Stereo-based approaches that use two (or more) cameras are popular, however they require (a) synchronisation of the cameras; (b) elaborate calibration procedures (e.g. [8, 11–13]); (c) collection of large amounts of data, particularly when using high frame rates; and (d) substantial post-processing that entails frame- by-frame tracking of individual features in all of the video sequences, and establishing the correct correspondences between these features across the video sequences (e.g. [14]). This is particularly complicated when tracking highly deformable objects, such as flying birds.

Vicon-based stereo trackers simplify the problem of feature tracking by using special reflective markers or photodiodes attached to the animal (e.g. [5, 6, 15, 16]). However, these markers can potentially disturb natural movement and behaviour, especially when used on small animals.

A novel recent approach uses structured light illumination produced by a laser system in combination a high-speed video camera to reconstruct the wing kinematics of a freely flying parrotlet at 3200 frames/second [10]. However, this impressive capability comes at the cost of

excellence (MVS): ARC Discovery Grant (DP 110103277) https://www.arc.gov.au/grants/discovery-program/discovery-projects (MVS): Human Frontiers in Science Grant (RGP0003/2013) https://www.hfsp.org/funding/hfsp-funding/research-grants (MVS): Queensland Smart State Premier's Fellowship https://advance.qld.gov.au/assets/includes/docs/research-fellowships-guidelines.pdf (MVS): ARC Distinguished Outstanding Researcher Award (DP140100914) https://www.arc.gov.au/grants/discovery-program/discovery-projects The funders had no role in study design, data collection and analysis, decision to publish, or preparation of the manuscript.

**Competing interests:** The authors have declared that no competing interests exist.

some complexity and works best if the bird possesses a highly reflective plumage of a single colour (preferably white).

GPS-based tracking methods (e.g. [17]) are useful for mapping long-range flights of birds, for example, but are not feasible in indoor laboratory settings, where GPS signals are typically unavailable or do not provide sufficiently accurate positioning. Furthermore, they require the animal to carry a GPS receiver, which can affect the flight of a small animal.

A simple technique for reconstructing 3D flight trajectories of insects from a single overhead video camera involves tracking the position of the insect as well as the shadow that it casts on the ground (e.g. [18, 19]). However, this technique requires the presence of the unobscured sun in the sky, or a strong artificial indoor light, which in itself could affect the animal's behaviour. (The latter problem could be overcome, in principle, by using an infrared source of light and an infrared-sensitive camera).

This paper presents a simple, inexpensive, compact, field-deployable technique for reconstructing the flight trajectories of birds in 3D, using a single video camera. The procedure for calibrating the camera is uncomplicated and is an exercise that needs to be carried out only once in the lifetime of the lens/camera combination, irrespective of where the system is used in subsequent applications.

The system was used in an earlier study of bird flight [20], but that paper provided only a cursory description of the technique. This paper provides a comprehensive description of an improved and extended version of the underlying technique and procedure, which will enable its application to other laboratory and field studies of bird flight.

## 2. Methodology

### 2.1 Derivation of basic method

The birds used in this study were Budgerigars (*Melopsittacus undulatus*). Information on their training and care is provided in the Section A in S1 File.

Our method uses a single, downward-looking camera positioned at the ceiling of the experimental arena in which the birds are filmed. The camera must have a field of view that is large enough to cover the entire volume of space within which the bird's flight trajectories are to be reconstructed.

Essentially, the approach involves combining knowledge of the bird's wingspan (which provides a scale factor that determines the absolute distance of the bird from the camera) with a calibration of the camera that uses a grid of known geometry drawn on the floor. This calibration provides a means of accounting for all of the imaging distortions that are introduced by the wide-angle optics of the camera lens. In our initial derivation, we assume that the bird does not display a significant amount of roll during its tracked flight. In other words, the two wingtips are in the horizonal plane (or approximately so) when the wings are fully extended. In the Section C in S1 File we derive and validate a method that accounts for the effects of roll, and also measures the roll angle.

A square grid of known mesh dimensions is laid out on the floor. The 2D locations (X,Y) of each of the intersection points are therefore known. Fig 1 illustrates, schematically, a camera view of the grid on the floor, and of a bird in flight above it, as imaged in a video frame in which the wings are fully extended.

In general, the images of the grid cells will not be square, but distorted by the non-linear off-axis imaging produced by the wide-angle lens, as shown in the real image of Fig 2.

The intersection points of the grid in the camera image are digitised (manually, or by using specially developed image analysis software), and their pixel locations are recorded. Thus, each grid location $(X_i, Y_i)$ on the floor is tagged with its corresponding pixel co-ordinates $(px_i, py_i)$

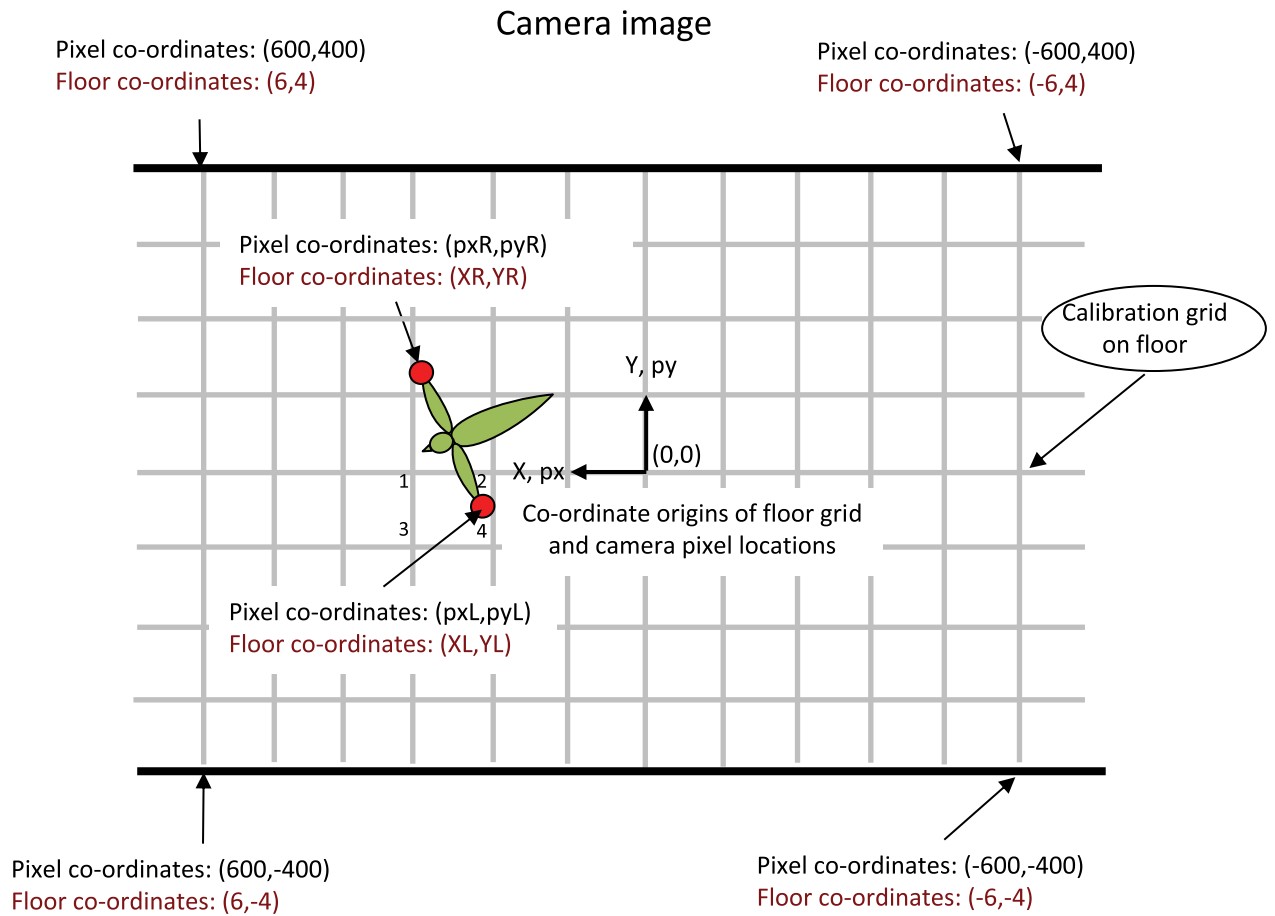

**Fig 1. Schematic view of image of the flight chamber.** The view is from an overhead video camera, showing the calibration grid on the floor, and the instantaneous position of a bird with its wings extended. The origin of the pixel co-ordinates is taken to be the center of the image, i.e., corresponding to the direction of the camera's optic axis. The origin of the calibration grid is taken to be the point directly beneath the camera, i.e., the position where the optic axis of the camera intersects the floor.

in the image. These data are used to compute a function that characterises a two-dimensional mapping between the grid locations on the floor and their corresponding pixel co-ordinates in the image. We note that the calibration grid does not need to be on the floor: It can be at any distance from the camera, but this distance must be known, and the plane of the grid must be perpendicular to the camera's optic axis.

Video footage of a bird flying in the chamber, as captured by the overhead camera, is then analysed to reconstruct the bird's 3D flight trajectory, as described below. Two examples of such footage are provided in the S1 and S2 Videos. The positions of the wingtips are digitised in every frame in which the wings are fully extended, i.e., when the distance between the wing-tips attains its maximum value (equal to the wingspan) in the video image. In the Budgerigar this occurs once during each wingbeat cycle, roughly halfway through the downstroke. We call these frames *Wex* frames, and denote the pixel co-ordinates of the wingtips in these frames by (pxL,pyL) (left wingtip) and (pxR,pyR) (right wingtip). The projected locations of the two wingtips on the floor are determined by using the mapping function, illustrated in Fig 2, to carry out an interpolation. Essentially, the projected location of this wingtip on the floor is obtained by computing the position of the point on the floor that has the same location,

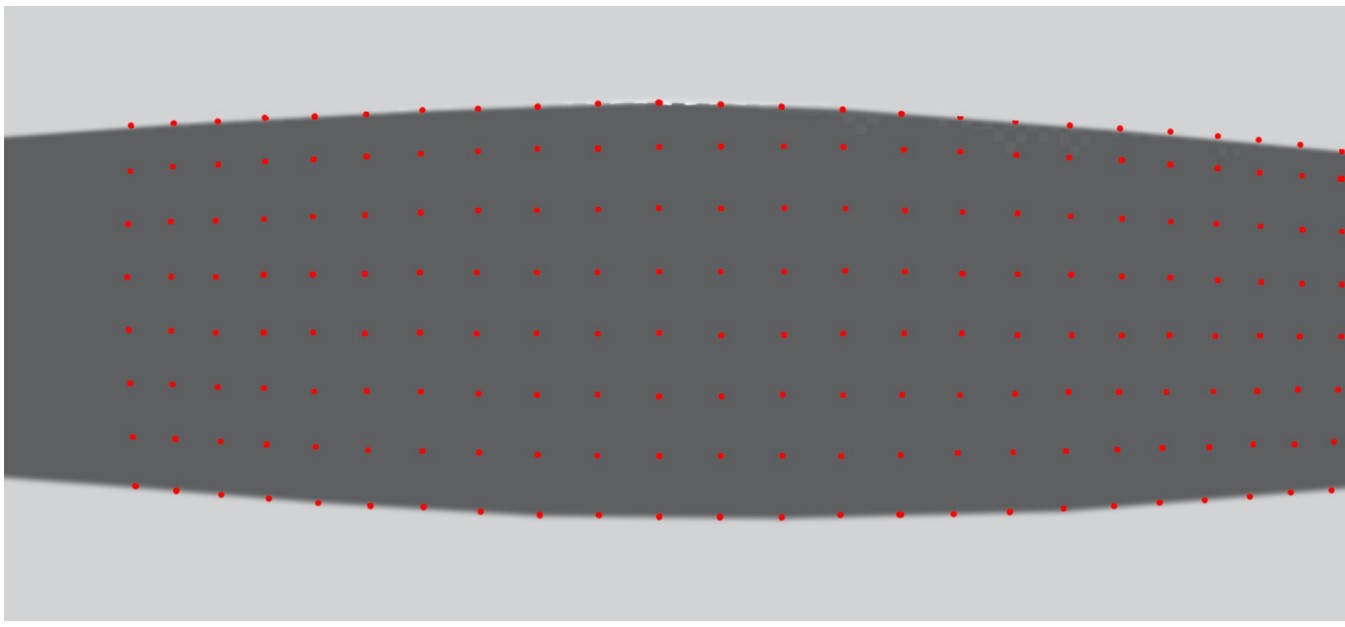

**Fig 2. Calibration grid.** Camera view of the calibration grid on the floor (red points).

relative to its four surrounding grid points, as does the position of the wingtip (in image pixel co-ordinates) in relation to the positions of the four surrounding grid locations (in image pixel co-ordinates). Thus, in the case of the left wing tip, for example, this computation effectively uses the locations of the four grid points 1,2, 3 and 4 (see Fig 1) with locations (X1,Y1), (X2, Y2), (X3,Y3) and (X4,Y4) on the floor, and their corresponding image pixel co-ordinates (px1, py1), (px2,py2), (px3,py3) and (px4,py4) respectively, to interpolate the projected position of the pixel co-ordinate (pxL,pyL) on the floor. A similar procedure is used to project the position of the right wingtip (pxR,pyR) on the floor. The construction of the two-dimensional mapping function, and the interpolation are accomplished by using the Matlab function *TriScatteredInterp*. (Equivalent customized codes could be written in any language.)

The total size of the grid must be large enough to capture the projected images of the wingtips on the floor over the entire flight trajectory of the bird. If the camera's imaging system approximates that of a pinhole camera (with no optical distortions) the grid cells can be large, because the linearly interpolated projections of the wingtips on the floor would be 100% accurate. But if optical distortions are present, the accuracy of the interpolated positions can be improved by increasing the density of the grid. This density can be arbitrarily large—it is limited only by the precision and effort required to construct a dense grid. However, due to the unavoidable errors associated with pixel digitization, it would probably not be beneficial for the image of an individual square to be smaller than about 5x5 pixels.

Once the positions of the two wingtips have been projected on to the floor, this information can be used to determine the instantaneous position of the bird in three dimensions, as illustrated in Fig 3. In this figure the 3D positions of the left and right wingtips are denoted by M, with co-ordinates (xL,yL,z), and N, with co-ordinates (xR,yR,z), respectively. Their projected points on the floor are denoted by C, with co-ordinates (XL,YL,0), and D, with co-ordinates (XR,YR,0), respectively.

The height of the bird above the floor is established by determining the ratio between the known wingspan of the bird (w), and the projection of its wingspan on the floor, which we

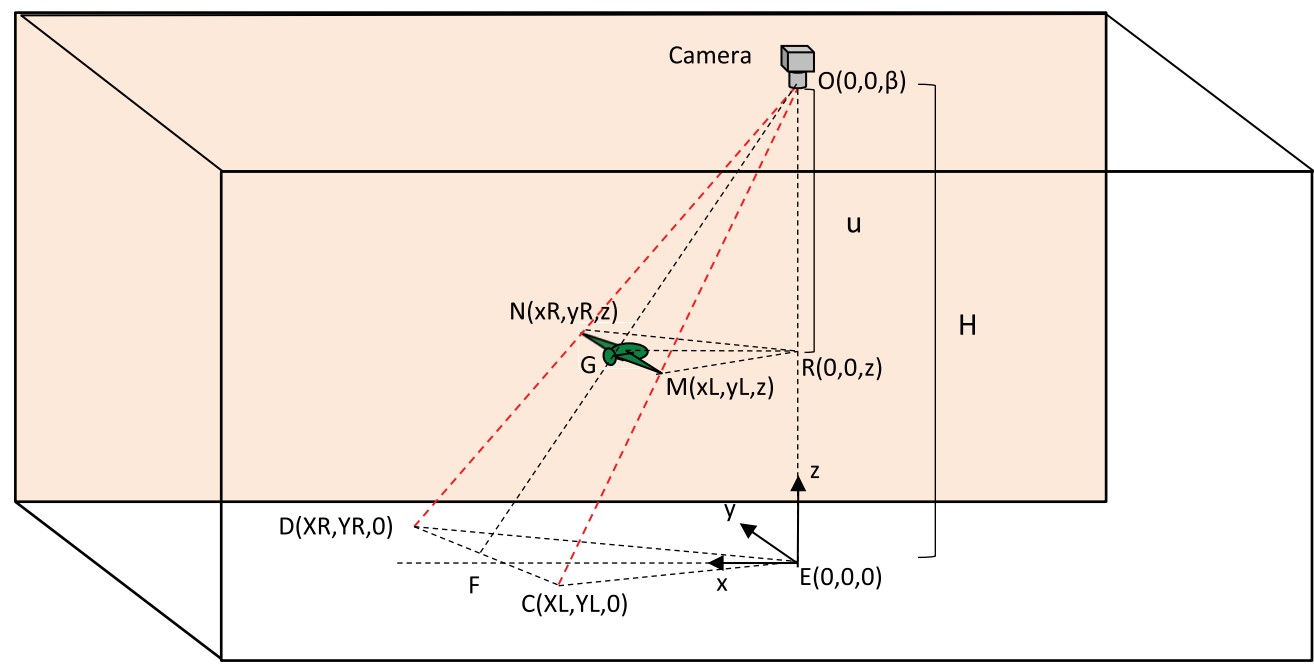

**Fig 3. Experimental chamber.** Schematic view of experimental chamber, showing the variables used for computing the instantaneous 3D position of the bird and its wingtips. E is the point on the floor that is directly beneath the camera, i.e., the point where the camera's optic axis intersects the floor.

denote by W. W, which is equal to the distance between points C and D in Fig 3, is given by

$$W = \sqrt{\left[(XL - XR)^2 + (YL - YR)^2\right]} \tag{1}$$

We denote the ratio (W/w) by Q.

From the geometrical similarity of the triangles OCD and OMN, and triangles OEF and ORG, we can write

$$Q = \frac{W}{w} = \frac{H}{u} \tag{2}$$

where H is the height of the ceiling (assumed to be known), and $u$ is the distance of the bird below the ceiling. The height $h$ of the bird above the floor, equal to (H−u), is then computed from (2) as

$$h = H\frac{(Q - 1)}{Q} \tag{3}$$

$h$ is the height of the two wingtips above the floor. The (x,y) co-ordinates of the left and right wingtips can also be computed from the wingspan ratio Q as follows.

From the similarity of triangles ODF and ONG, and OEF and ORG, we have:

$$\frac{CF}{MG} = \frac{OE}{OR} = \frac{H}{u} = Q \tag{4}$$

which can be rewritten as

$$MG = \frac{CF}{Q} \tag{5}$$

This implies that the (x,y) position co-ordinates of the left wingtip are given by

$$xL = \frac{XL}{Q}, \text{ and } yL = \frac{YL}{Q} \tag{6}$$

and the (x,y) position co-ordinates of the right wingtip are

$$xR = \frac{XR}{Q}, \text{ and } yR = \frac{YR}{Q} \tag{7}$$

Thus, the 3D position co-ordinates of the left and right wing tips are $(xL,yL,h)$ and $(xR,yR, h)$. If we assume that the centre of the bird (the approximate position of its centre of gravity) is located midway between the extended wingtips (i.e., approximately at the thorax), then the 3D co-ordinates of the centre of the bird $(xc,yc,zc)$ (which we shall henceforth refer to as the thorax) can be computed as

$$(xc, yc, zc) = \left[\frac{(xL + xR)}{2}, \frac{(yL + yR)}{2}, h\right] \tag{8}$$

However, computing the centre of the bird in this way is valid only at the instants when the wings are fully extended. At other times the wings would be pointing either forward or backward, and this calculation would yield an incorrect result. Another approach would be to track the position of the head. During flight, the head is the most stable part of the bird's anatomy- it maintains a horizontal orientation that is largely independent of the pitch and roll attitude of the body [21–23]. It is also a highly visible part of the bird that can be tracked reliably—either manually through frame-by-frame digitisation, or by software algorithms that employ relatively simple heuristics. Moreover, the head carries the bird's primary sense organs, including the eyes. Therefore, reconstructing the 3D trajectory of the head can be useful for determining the visual stimuli that the bird experiences during its flight.

The 3D position co-ordinates of the head can be calculated for each frame as follows. The pixel co-ordinates of the head are determined in every frame (either through manual digitisation or an automated tracking algorithm). The head pixel co-ordinates are projected on to the floor, using the same interpolation procedure that was applied to the wingtips. We denote the floor co-ordinates of the head by (XH,YH) (not shown in Fig 3). Then, using the same geometrical reasoning as above, the (x,y) position co-ordinates of the head are given by

$$xH = \frac{XH}{Q}, \text{ and } yH = \frac{YH}{Q} \tag{9}$$

and the full 3D co-ordinates of the head are given by

$$(xH, yH, zH) = \left[\frac{XH}{Q}, \frac{YH}{Q}, h\right] \tag{10}$$

We note that the height of the head ($h$) is directly calculable only in the frames in which the wings are fully extended, because the bird's wingspan is the known metric that enables determination of the height. The heights in other frames are estimated through temporal interpolation, assuming that the height varies approximately linearly between successive wing extensions. This is a reasonable assumption for most birds—typically, the height of flight varies

slowly and smoothly across several wingbeat cycles. However, the X and Y coordinates of the bird in 3D ($xH,yH$) are determined independently for each frame of the video sequence, based on the digitised pixel co-ordinates of the head in each frame, and the temporally interpolated height for that frame. Thus, while the height of the head (h) is temporally interpolated between wing extensions, the (X,Y) co-ordinates of the head ($xH,yH$) can be calculated independently for each frame, based on the pixel co-ordinates of the head in that frame. The height of the thorax can be estimated through a similar calculation, by defining the thorax either as the midpoint between the extended wingtips in the image, or the midpoint between the wing bases in the image.

In summary, our method delivers a sample of the bird's height at every frame in which the wings are extended. These samples are interpolated in time to obtain a height profile of the head for the entire video sequence. This height profile is then used in combination with the pixel co-ordinates of the head in each frame to obtain the 3D co-ordinates of the head for each frame of the video sequence. The 3D trajectory of the thorax (defined as the mid-point between the extended wingtips) can also be reconstructed as described above.

In Budgerigars, the wings are fully outstretched only once during each wingbeat cycle–roughly halfway through the downstroke, as we have noted above. This also appears to be the case in pigeons and magpies [24]. It is possible that in certain other species, which move their wings in the same plane during the upstroke and the downstroke, without folding them, there are two *Wex* frames per wingbeat cycle—one occurring during the upstroke, the other during the downstroke. In such cases we can obtain two height estimates per wingbeat cycle, and therefore reconstruct the height profile at twice the temporal resolution.

In the above analysis, we have assumed that the head of the bird is at the same height as that of its extended wingtips. If the head is at a different height—as may be evinced from prior knowledge or from side-view images of bird flight in wind tunnels—this known height offset can be added to the wingtip height to obtain the true height of the head.

## 2.2 Procedural steps

Based on the theory described above, the step-by-step procedure for reconstructing the 3D trajectory of the head of a bird from a video sequence captured by a single overhead camera can be described as follows:

i.  Construct the floor grid and acquire an image of the grid from the video camera. An example is shown in Fig 2. The grid is used only once for the camera calibration, and does not need to be present in the experiments.

ii.  Digitise the pixel co-ordinates of the grid locations in the camera image, to obtain a one-to-one mapping between the real co-ordinates of the grid locations on the floor and their corresponding pixel coordinates in the image.

iii.  Acquire knowledge of the bird's wingspan, either from published data for the species, or, preferably, from direct measurement of the actual individual (because the wingspan can vary slightly across individuals due to age and other factors).

iv.  Acquire video footage of the bird during flight in the chamber

v.  Select the frames in the video sequence in which the wings are fully extended. The selection can be done either manually, or through custom-written software. The wing-extension frames are denoted by *Wex*.

vi.  Digitise the pixel positions of the left and right wingtips of the bird in each of the *Wex* frames, as shown in the illustrative example of Fig 4.

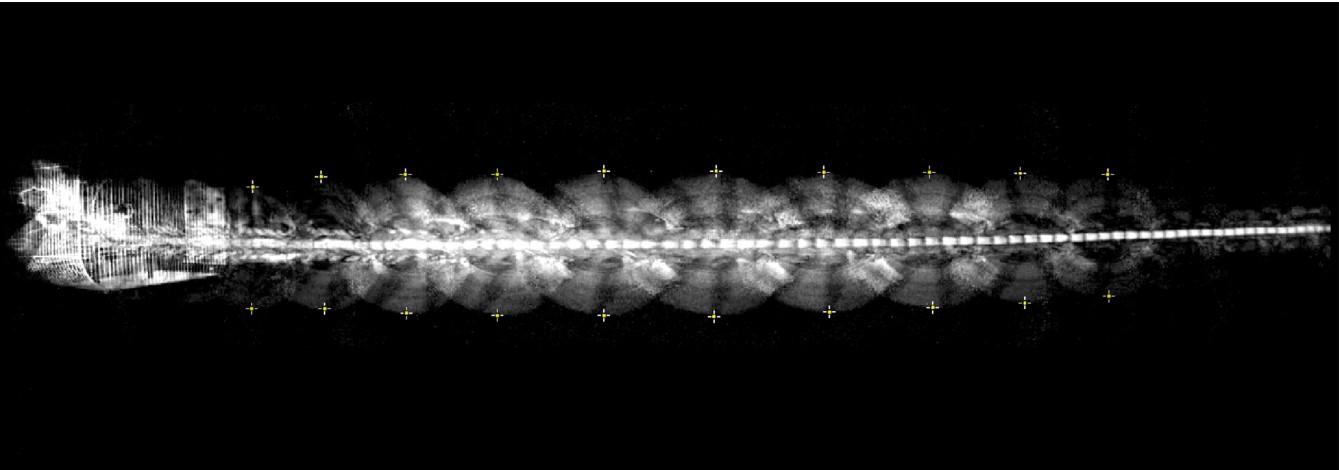

**Fig 4. Video of a flight.** Example of a video sequence showing superimposed images of the bird in successive frames. Successive wing extensions are marked by the crosses.

vii. Determine the height of the thorax (midpoint between the extended wingtips) or of the head in each of the *Wex* frames from Eqs (1–3).

viii. Obtain the height profile of the thorax (and/or the head) for the entire video sequence by temporally interpolating the heights calculated for the *Wex* frames.

ix. Digitise the pixel position of the thorax/head in each frame of the video sequence.

x. Compute the 3D position of the thorax/head for each frame from Eqs (9) and (10).

### 2.3 Test of accuracy

The precision of the 3D trajectory reconstruction procedure was evaluated by placing a small test target at 44 different, known 3D locations within the tunnel, of which 39 were within the boundary of the grid. The test target was a model bird with a calibrated wingspan of 30 cm. The head was assumed to be midway between the wingtips, and at the same height as the wingtips. This assumption does not affect the generality of the results, as discussed above. The standard deviations of the errors along the x, y and H directions were 2.1 cm (X), 0.6 cm (Y) and 2.6 cm (H). A detailed compilation of the errors is given in S1 Table in S1 File.

### 2.4 Ethics statement

All experiments were carried out in accordance with the Australian Law on the protection and welfare of laboratory animals and the approval of the Animal Experimentation Ethics Committees of the University of Queensland, Brisbane, Australia (Animal Ethics Approval Certificate QBI/075/18).

## 3. Results

### 3.1 Examples of flight tracking and reconstruction

Here we show some examples of reconstruction of 3D trajectories of flights of Budgerigars through an indoor tunnel, of dimensions of dimensions 7.28 m (length) x 1.36 m (width) x 2.44 m (height). The birds were trained to fly from a perch at one end of the tunnel to a bird cage at the other. A downward-facing video camera, placed at the centre of the ceiling of the

tunnel, was used to film the flights and reconstruct the trajectories in 3D. A grid, of check size 20 cm x 20 cm (as in Fig 2), was drawn on the floor to calibrate the camera using the procedure described above. The reconstructed 3D trajectories do not include the take-off and landing phases of the flight. They only show a section of the trajectory within a window that extends from about 1.75 m ahead of the camera aperture to about 0.25 m behind it, which could be regarded as a 'cruise' phase where the bird has completed take-off and not yet commenced landing.

Flights through the tunnel were filmed with the tunnel being either empty (devoid of any obstacles) or carrying a narrow, vertically oriented aperture (a slit) at the halfway point, through which the birds had to fly to get to the other end. To prevent injuries to the birds, the aperture was created by suspending two cloth panels that reached from the ceiling to the floor. Two aperture widths were tested: In one set of tests, the aperture was 5cm wider than the bird's wingspan; in the other set, the aperture was 5cm narrower than the bird's wingspan. It has been shown in earlier studies [20, 25] that Budgerigars are acutely aware of their wingspan: when negotiating a narrow aperture, they fold their wings back briefly only when the aperture is narrower than their wingspan, and fly through without interrupting their wingbeat cycle if the aperture is wider than their wingspan.

A plan view of a reconstructed flight is shown in Fig 5. In this example, the bird (*Four*) has a wingspan of 29 cm and it flies through a 34 cm aperture, which is 5 cm wider than the wingspan. The figure shows the (X,Y) positions of the two wingtips at the time of each wing extension, the thorax (defined as the midpoint of the line joining the extended wingtips), and the

Fig 5. Plan view of a reconstructed flight of bird Four. In this example, the wingspan of the bird is 29 cm and it flies through a 34 cm aperture, which is 5 cm wider than the wingspan. Details in text. The red circles show the wingtip positions at the time of each wing extension, the black circles show the inferred position of the thorax at these instants, and the blue asterisks depict the position of the head at these instants. The red lines show the wing extension trajectories interpolated between wing extensions. The arrow in this and other figures shows the direction of flight. In this figure and subsequent figures (Figs 6–8 and S1-S6 Figs in S1 File), the y axis ("Tunnel width") refers to the position (in meters) along the width of the tunnel (corresponding to the y axis in Fig 3).

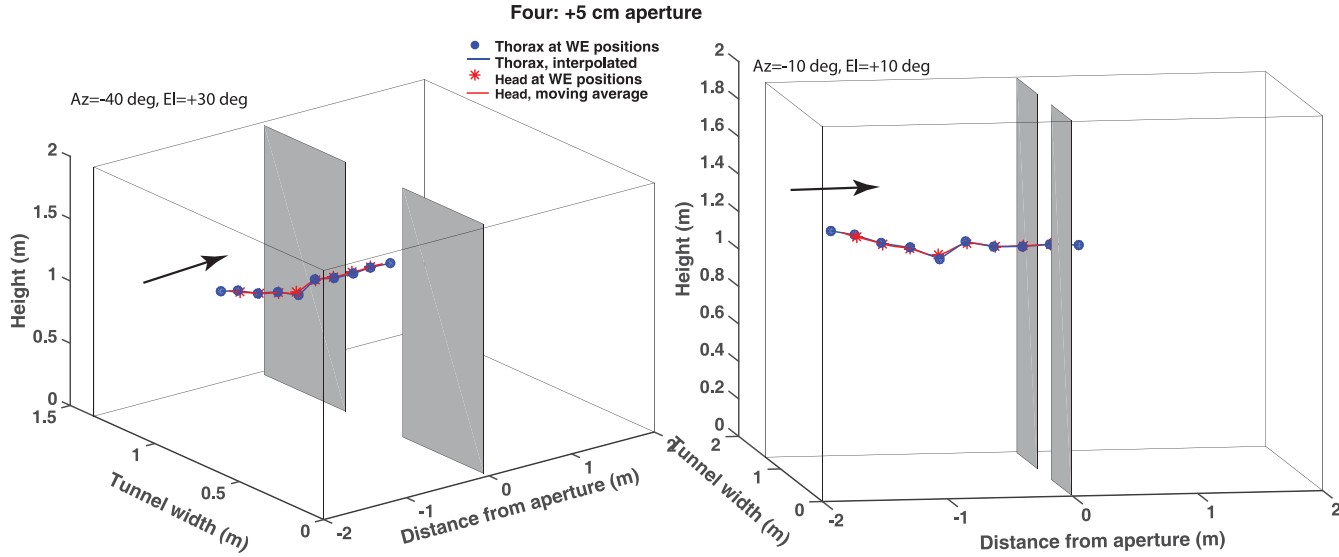

**Fig 6. Two 3D views of the trajectory shown in Fig 5.** Bird Four flies through an aperture that is 5 cm wider than its wingspan. The blue circles show the inferred position of the thorax at the time of each wing extension, the blue lines show the linearly interpolated thorax positions between successive wing extensions, and the red asterisks show the head position at the time of each wing extension. The image coordinates of the head, which were digitized in every video frame, were used to calculate the 3D trajectory of the head in every frame, as described in the text. The red curve shows the resulting 3D trajectory of the head during the entire video sequence, after smoothing by a 9-point centered rectangular moving average filter. Left panel: View from -40 deg azimuth, 30 deg elevation. Right panel: Near-lateral view from -10 deg azimuth, 10 deg elevation.

position of the head. It is evident that the bird flies through the aperture without interrupting its wingbeat cycle, as the wingbeat extensions are equally spaced.

This is also clear from Fig 6, which shows two 3D views of the same flight trajectory, where the blue circles represent the position of the thorax at each wing extension and the red curve shows the reconstructed 3D position of the head for every frame, as described in the text above and in the legend. The lateral view of the trajectory (Fig 6, right hand panel) shows that the bird maintains its height while passing through the aperture, because the wingbeat cycle is not interrupted.

Fig 7 shows two 3D views of a trajectory of the same bird during flight through an aperture that is 5 cm narrower than its wingspan. Here it is clear that the wingbeat cycle is interrupted when the bird passes through the aperture–the distance between successive wing extensions is dramatically larger during the passage. This is also evident from the lateral view of the trajectory (Fig 7, right hand panel), which shows that the bird loses altitude while passing through the aperture, because the wingbeat cycle is interrupted.

S5 Fig in S1 File (Section B, in S1 File) shows two 3D views of a trajectory of the same bird during flight through the tunnel when there is no aperture. In this case–as in Fig 6 - the wingbeat cycle is not interrupted anywhere in the flight. This is also clear from the lateral view of the trajectory (S5 Fig in S1 File, right hand panel), which shows that the bird maintains a regular wingbeat cycle and does not lose altitude abruptly anywhere along the trajectory.

It is clear from Figs 5–7 and S4 Fig in S1 File that bird *Four* interrupts its wingbeat cycle only when it confronts an aperture that is narrower than its wingspan, and not when the aperture is wider than the wingspan or is not present in the tunnel. A loss of altitude occurs only when the wingbeat cycle in interrupted, and not otherwise.

Fig 8 shows plan views of the reconstructed 3D trajectories of the head for the three conditions. In each case, the asterisks mark the locations of the head at the times of full wing

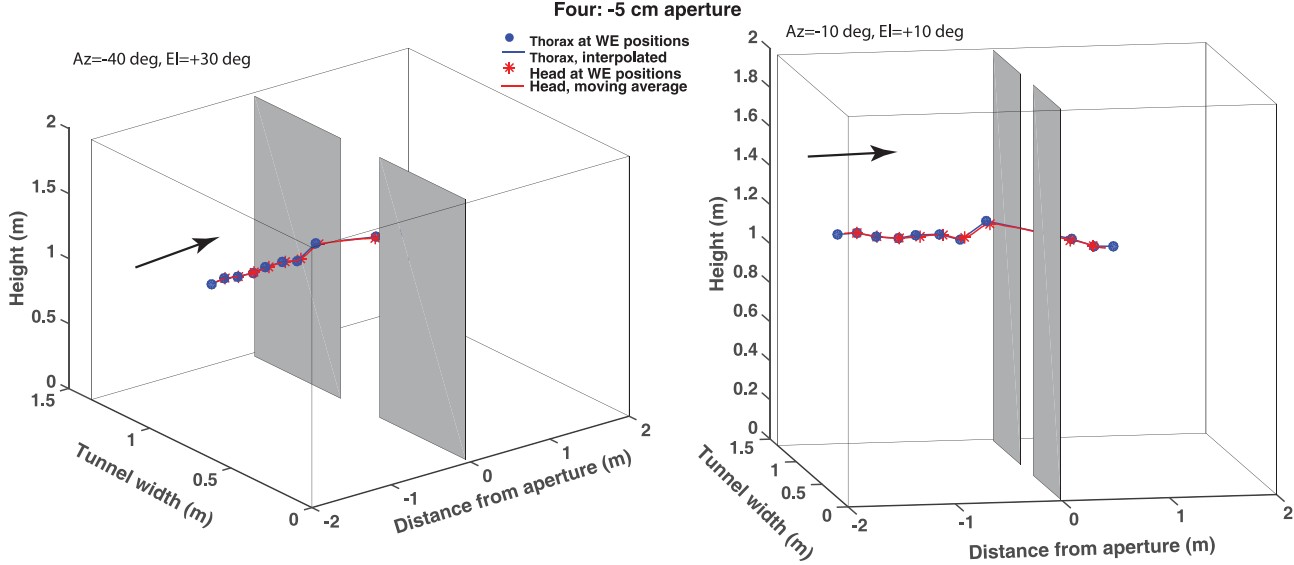

**Fig 7. Two 3D views of a trajectory of bird Four during flight through an aperture that is 5 cm narrower than its wingspan.** Details are as in Fig 6.

extension. Other details are given in the figure legend. In the case of the narrow aperture (red track), the bird temporarily interrupts its wingbeat cycle while passing through the aperture. The final wing extension prior to passing the aperture occurs at a point approximately 0.35 m ahead of the aperture. The wingbeat cycle resumes after passage through the aperture, with the first wing extension occurring at a point approximately 0.5m beyond the aperture. In the wide aperture and the no aperture conditions, the wingbeat cycle continues uninterrupted throughout the flight. These observations are in agreement with those of [25], who report an exquisite ability of these birds to gauge the width of oncoming passages in relation to their wingspan. However, their study only recorded the frequency and timing of wing closures, and did not reconstruct the birds' trajectories in 3D.

Fig 9 shows reconstructed profiles of the forward flight speed (speed along the X axis of the tunnel) for the flights of bird *Four* in the narrow aperture, wide aperture and no-aperture conditions. These profiles were constructed using three different procedures, the details of which are described in the legend. The three procedures yield consistent results. The principal observation is that the forward speed is more or less constant throughout the flight and is independent of the flight condition, as observed in [20]. Interestingly, the interruption of the wingbeat cycle during the flight through the narrow aperture does not significantly reduce the forward speed.

In the Supplementary Information (S1-S4, S6, S7 Figs in S1 File) we show results for another bird (*Nemo*), corresponding those shown above for bird *Four*.

## 3.2 Accounting for the effects of body roll: Extended calculation

In the analysis so far, we have assumed that the birds are not rolling during flight, i.e., that the wingtips are in the horizontal plane when they are fully extended. This assumption is quite valid for the flights we have filmed and reconstructed: the birds displayed very little roll throughout their flight, as evinced by the fact that, when the wings were outstretched, the two wingtips were approximately equidistant from the head in all video frames. However, our theory can be extended to take body roll into account–when this is significant–and continue to

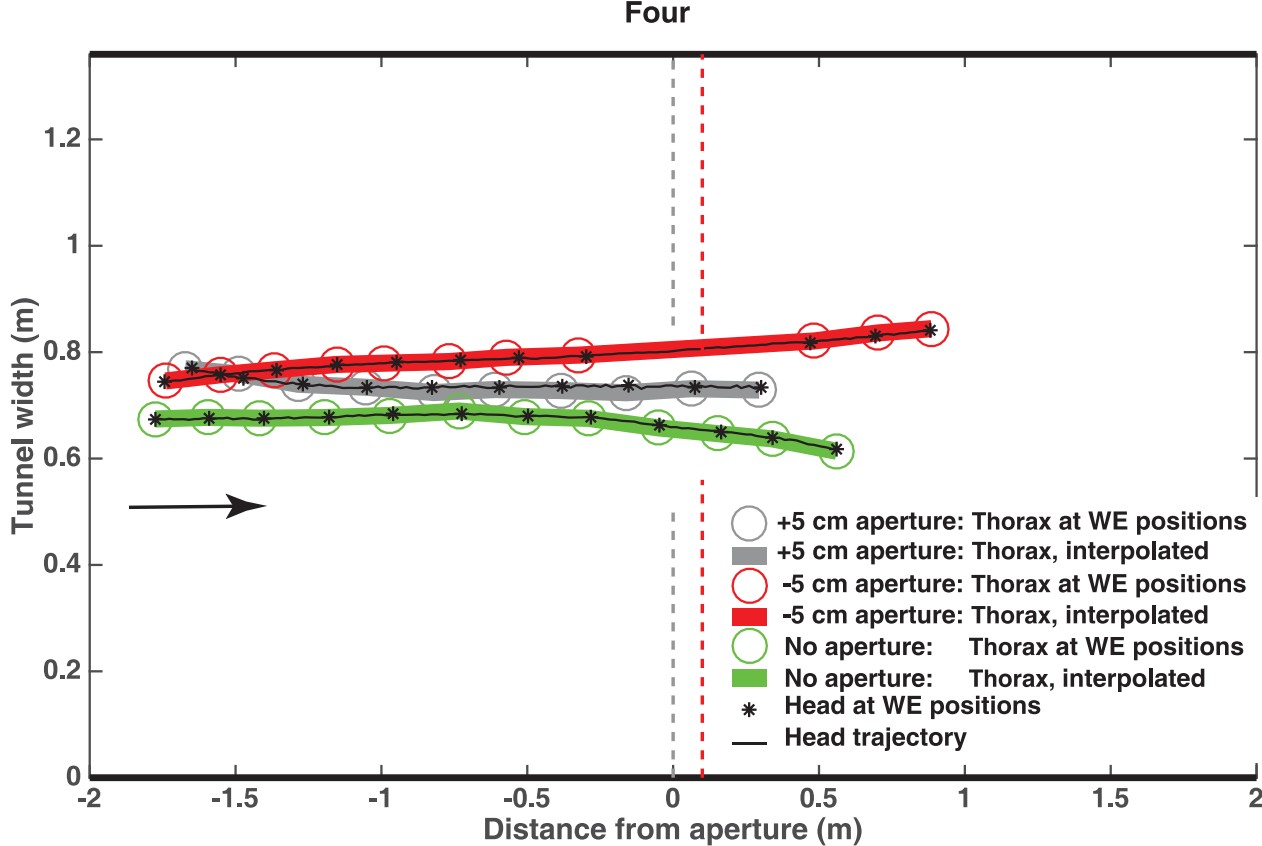

**Fig 8. Plan views of 3D trajectories.** Trajectories reconstructed for the narrow aperture condition (red), the wide aperture condition (grey) and the no aperture condition (green), for bird Four. In each case, the circles mark the locations of the thorax (defined as the mid-point of the line connecting the extended wing tips) at the time of each wing extension, the thick curves show the thorax trajectory interpolated from the thorax positions at these times, the asterisks mark the locations of the head at the times of the wing extensions, and the thin black curve through the asterisks shows the trajectory of the head, reconstructed from the digitized image co-ordinates of the head in each frame as explained in the text.

obtain accurate estimates of the 3D trajectories, as well as the roll angles. The essential elements of the procedure are described briefly below, and the complete general derivation is provided in the Section C in S1 File.

The basic principle underlying the calculation of the roll angle is outlined in Fig 10, which illustrates a simplified case in which the midpoint O between the extended wingtips is directly under the camera, i.e., in line with the camera's optic axis. (The more general case, in which the bird can be at any location, is dealt with in Section C of the S1 File). During a roll, the line connecting the extended wingtips is not horizontal. Therefore, in the camera image the angles $\phi 1$ and $\phi 2$ subtended by the two wings will not be equal, because the right wingtip (in this case) is higher than the left wingtip. These angles can be computed from the projections on the grid floor of the images of the two wingtips (R,L), and the projection of the point (O), which is the point on the bird that is midway between the extended wingtips. When the bird is not rolling the extended wingtips will lie in a horizontal plane, and the image of O in the camera will be midway between the images of the extended wingtips, because $\phi 1 = \phi 2$. However, when the bird is rolling, $\phi 1 \neq \phi 2$ and the image of O will not be midway between the images of the extended wingtips. $\phi 1$ and $\phi 2$ can be measured and used to calculate the roll angle. In the camera image, O is determined as the point where the straight line connecting the wingtips

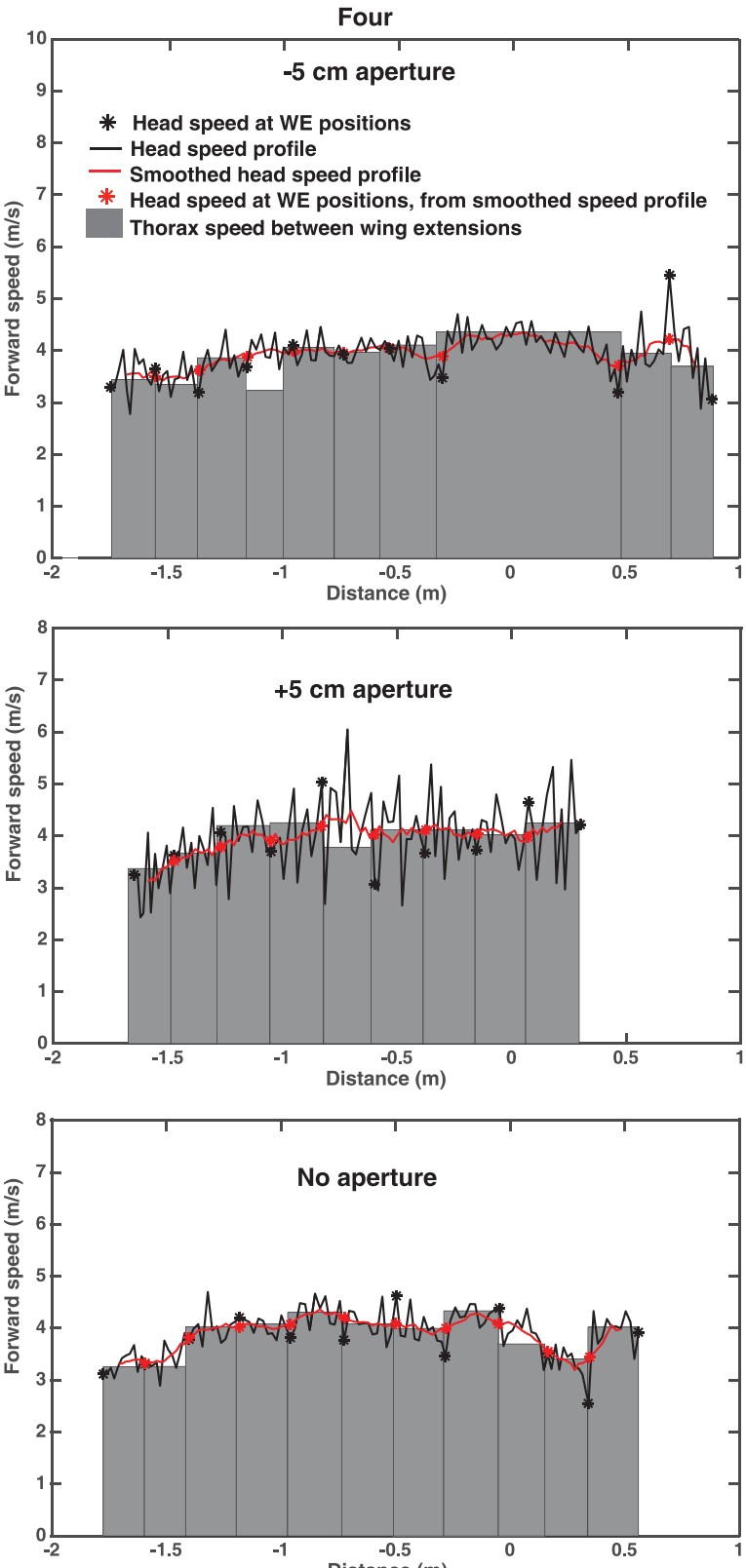

**Fig 9. Flight speed profiles.** Forward speed profile of bird Four during flight through the narrow aperture (top panel), the wide aperture (middle panel), and the empty tunnel (bottom panel). In each case, the black curve shows the speed profile of the head, computed from the frame-to-frame X positions of the head. The black asterisks denote the speeds at the head positions corresponding to the wing extensions, and the red curve and asterisks depict the result of smoothing the speed profile using a 9-point centered rectangular moving average filter. The edges of the grey bars denote the successive X positions of the thorax (defined as the midpoint between the wingtips) at each wing extension, computed as described in the text. The height of each grey bar depicts the mean forward speed of the thorax between successive wing extensions, computed as the ratio of the X distance between successive edges, to the time interval between these edges. Because the wingbeat kinematics are not perfectly identical from one wingbeat cycle to the next, and the head could make small movements relative to the thorax, the spatial relationship between the head and the line joining the wingtips is not exactly the same at the point of each wing extension. As a result, the measured speed of the thorax (grey bars) can occasionally be noticeably different from that of the head (asterisks): e.g. fourth grey bar (top panel), and fifth grey bar (middle panel).

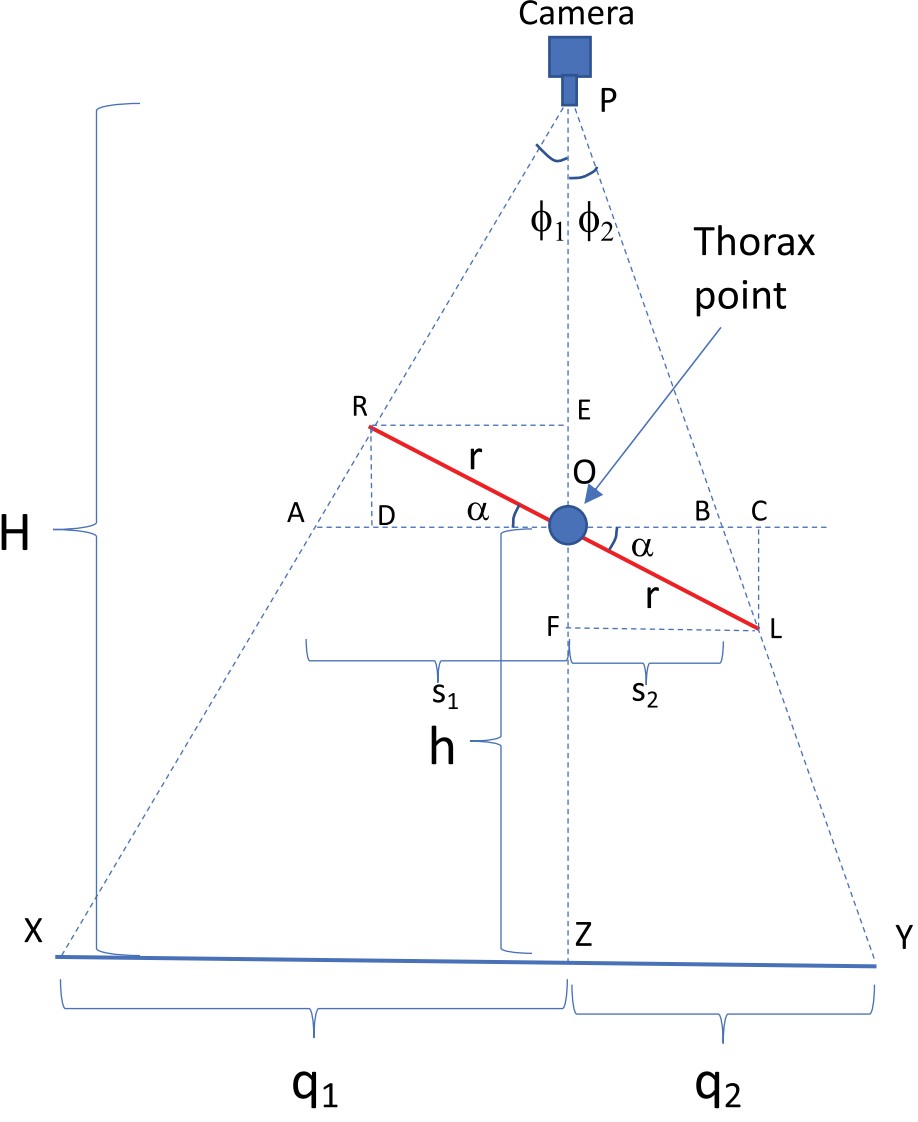

**Fig 10. Calculation of height and roll angle.** Illustration of calculation of bird height (h) and roll angle ($\alpha$) for a simplified case in which the midpoint of the wingspan is directly below the camera's optical axis. The known wingspan of the bird is 2r, and the camera is at a known height H above the floor.

intersects the longitudinal axis of the thorax (see Fig 13). We shall hereafter refer to O as the 'thorax point'. From the projected locations of R, L and O on the floor grid, the height of O above the ground (h) and the angle of roll ($\alpha$) can be calculated as shown below. This calculation distinguishes between changes in roll and changes in height, because it evaluates the distances of the projected wingtips *separately* for the left and right wings: these distances are not equal when the bird rolls.

From Fig 10 we have

$q_{1^\circ}$ = H tan $\phi_1$ and $q_{2^\circ}$ = H tan $\phi_2$, where

$$\emptyset_1 = \tan^{-1}\frac{q_1}{H} \tag{11}$$

$$\text{and } \emptyset_2 = \tan^{-1}\frac{q_2}{H} \tag{12}$$

so that

$$\frac{q_1}{q_2} = \frac{\tan\phi_1}{\tan\phi_2} \tag{13}$$

We can also write

$$s_1 = OD + DA = r\cos\alpha + r\sin\alpha\tan\phi_1 \tag{14}$$

and

$$s_2 = OC - BC = r\cos\alpha - r\sin\alpha\tan\phi_2 \tag{15}$$

Thus,

$$\frac{s_1}{s_2} = \frac{\cos\alpha + \sin\alpha\tan\phi_1}{\cos\alpha - \sin\alpha\tan\phi_2} \tag{16}$$

From triangle similarity we have

$$\frac{s_1}{s_2} = \frac{q_1}{q_2} \tag{17}$$

Equating (13) and (16) we obtain

$$\frac{\cos\alpha + \sin\alpha\tan\phi_1}{\cos\alpha - \sin\alpha\tan\phi_2} = \frac{\tan\phi_1}{\tan\phi_2} \tag{18}$$

which gives

$$2\sin\alpha\tan\phi_2\tan\phi_1 = (\tan\phi_1 - \tan\phi_2)\cos\alpha \tag{19}$$

or

$$2\tan\alpha = \frac{\tan\phi_1 - \tan\phi_2}{\tan\phi_1\tan\phi_2} = \frac{\sin(\emptyset_1 - \emptyset_2)}{2\sin\emptyset_1\sin\emptyset_2} \tag{20}$$

from which we obtain

$$\alpha = \tan^{-1}\left[\frac{\sin(\emptyset_1 - \emptyset_2)}{2\sin\emptyset_1\sin\emptyset_2}\right] \tag{21}$$

Thus, the roll angle $\alpha$ can be evaluated from (21), using (11) and (12) to calculate $\emptyset_1$ and $\emptyset_2$.

Now, considering the total length AB, we have

$$AB = AO + OB = s_1 + s_2 = r[2\cos \alpha + \sin \alpha(\tan \phi_1 - \tan \phi_2)] \tag{22}$$

Then, from triangle similarity we have

$$\frac{H - h}{H} = \frac{AB}{XY} = \frac{r[2\cos \alpha + \sin \alpha(\tan \phi_1 - \tan \phi_2)]}{XY} \tag{23}$$

from which we solve for $h$ to obtain

$$h = H\left[1 - \frac{r[2\cos \alpha + \sin \alpha(\tan \phi_1 - \tan \phi_2)]}{XY}\right] \tag{24}$$

Since X and Y are the projections of the right and left wingtips on the floor, the distance [XY] can be calculated, and $h$ can be evaluated using (24).

Note that Fig 10 illustrates a very simple case in which the thorax point of the bird is directly below the camera (in line with the camera's optical axis), for the purpose of conveying the basic approach to the calculation. In the Section C in S1 File we derive an extension of this calculation for a general case in which the bird can be at any 3D location. The extended calculation delivers the height of the bird, as well as the roll angle, irrespective of the position of the bird relative to the camera. Once the height is known the 3D co-ordinates of the head can computed as before, following the procedure described in Section 2.1 [Eq (10)].

### 3.3 Validation of extended calculation

We have validated the extended calculation, which accounts for body roll, by using a model bird in a scaled-down arena with an overhead camera and a calibration grid on the floor. A calibrated platform was used to position the model bird at 20 different 3D locations and flight directions, at various known roll angles and heights above the floor (Fig 11).

Four examples of the images captured by the overhead camera are shown in Fig 12.

Fig 13 compares the thorax heights and roll angles derived from the extended calculation with the true (ground truth) values, for each of the 20 configurations. It is evident that the extended analysis delivers accurate estimates of the height of the bird irrespective of body roll, as well as accurate estimates of the roll angle.

For the results shown in Fig 13, the mean and RMS errors in the calculated heights are 0.06 mm and 2.3 mm, respectively (0.03% and 1.3% of the wingspan, respectively), and the mean and RMS errors in the calculated roll angles are -0.4 deg and 1.7 deg, respectively.

While the real birds did not display significant body rolls during their flights in our experimental tunnel, the validation of our extended calculation using the model bird demonstrates that this procedure can be applied to calculate the true height (and the 3D coordinates of the thorax point and the head) of a bird even when its body is rolling, and to compute the roll angle.

## 4. Discussion

This study has described a simple, inexpensive method for reconstructing the flight trajectories of birds in 3D, using a single video camera. The advantages of the method are:

i.  The technique does not use a conventional stereo-based approach. Therefore, it does not require complex calibration procedures involving capturing views of a checkerboard at various positions and orientations, which does not always guarantee accurate localisation in all regions of the experimental space.

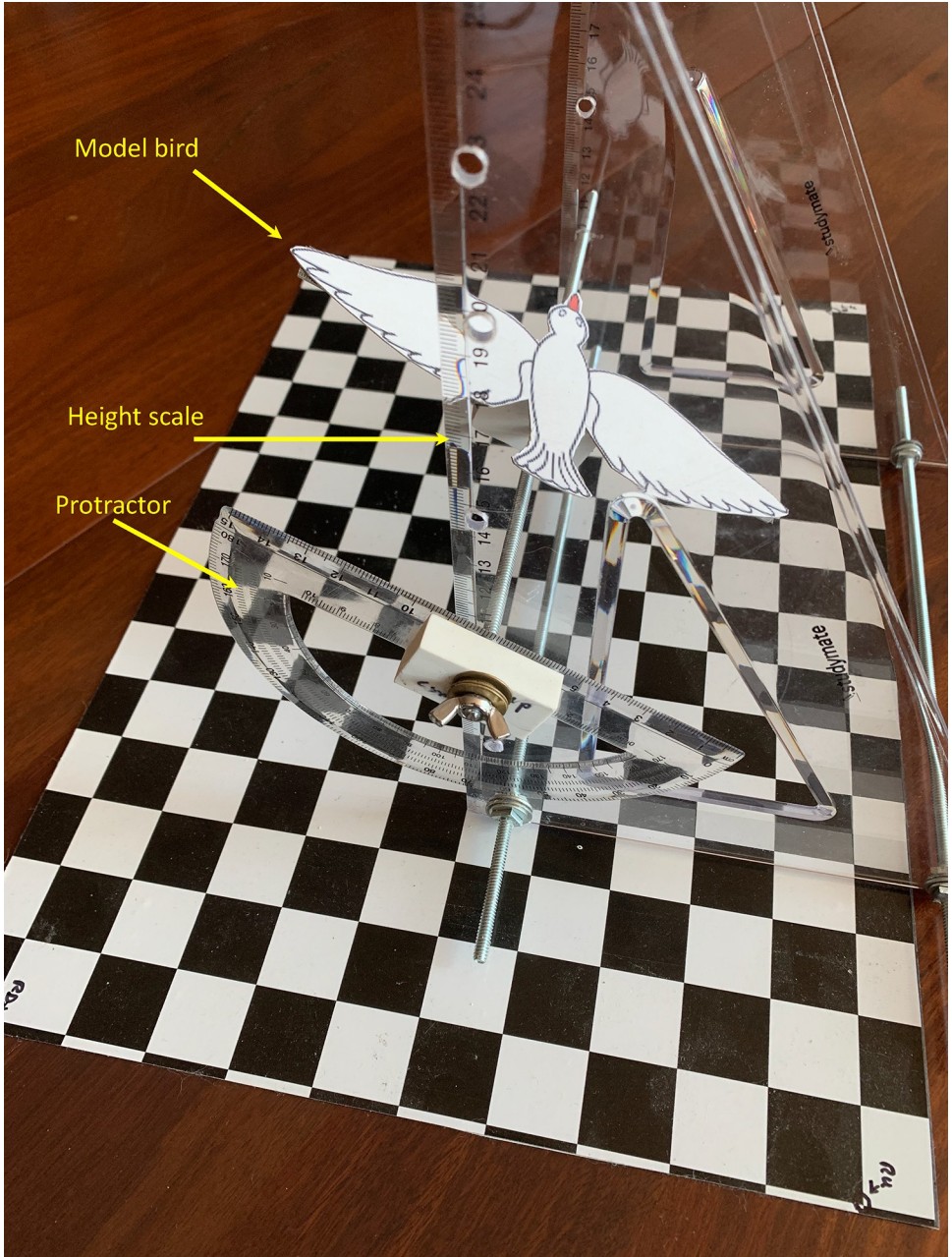

**Fig 11. Setup for validation of extended calculation, which accounts for body roll.** The model bird, with a wingspan of 180 mm, is mounted on a platform whose height and tilt (roll) can be set to calibrated values using the height scale and the protractor. The size of the floor grid is 408 x 260 mm, and the individual checks are 26.0 x 25.5 mm. The overhead camera is positioned above the center of the grid, with the nodal point of its lens at a height of 448 mm.

ii. The technique does not require feature correspondences to be determined across video frames from two or more cameras.

iii. The grid marker on the floor provides a calibration of the camera geometry and accounts for all of the distortions in the camera optics. There is no need to assume that the camera can be approximated by a pinhole camera, or by any other specific optical geometry. This

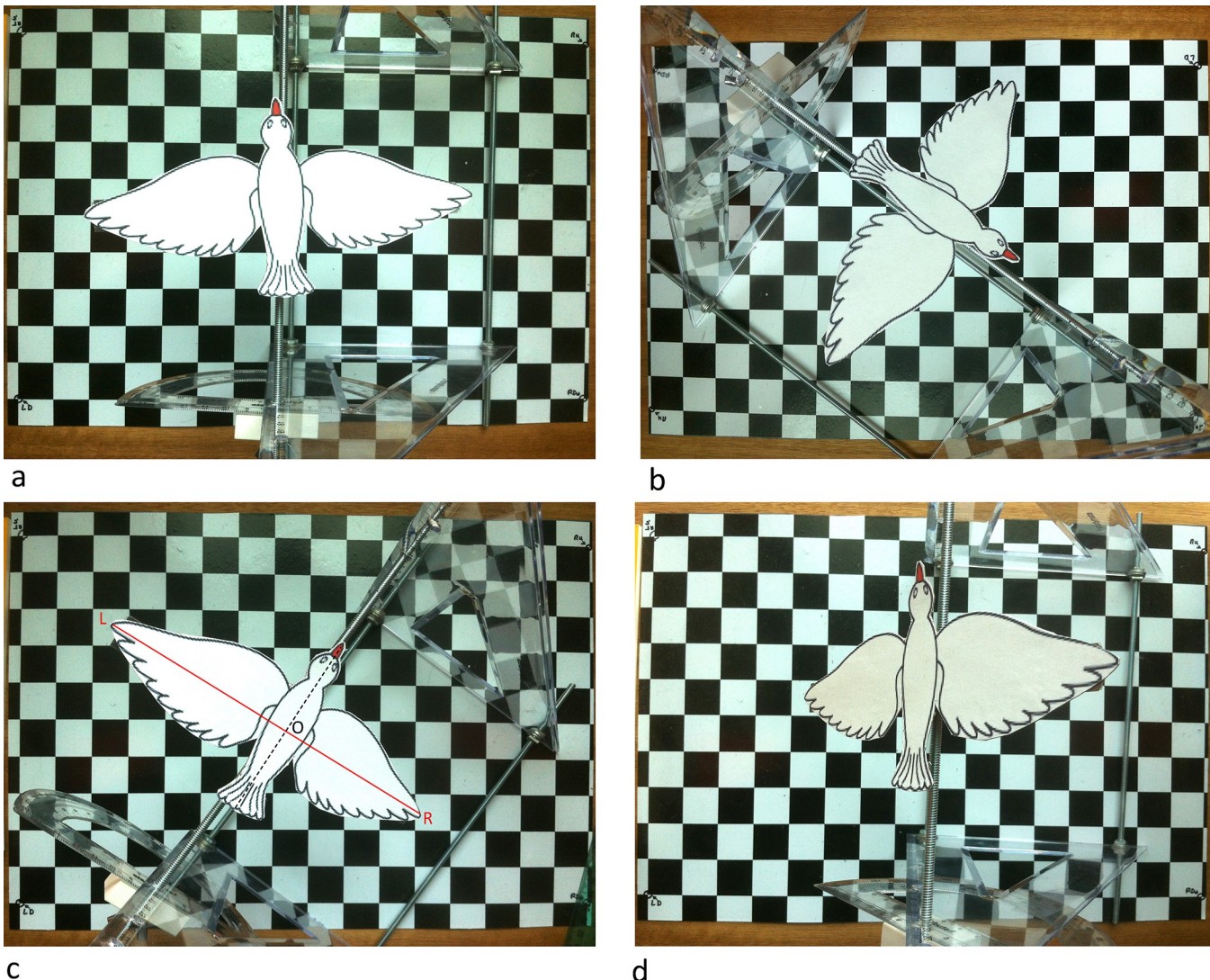

**Fig 12. Examples of images acquired by the overhead camera.** (a) Bird height 158mm, roll 0 deg; (b) Bird height 158mm, roll +26 deg; (c) Bird height 157mm, roll -22.5 deg; (d) Bird height 205 mm, roll +51 deg. Heights refer to the height of the thorax point O, as illustrated in (c), and in Fig 11, S7 and S8 Figs in S1 File. The roll angle is positive when the right wingtip is higher than the left wingtip, and vice versa. Note that it is not necessary for the thorax point to be directly below the camera (see, for example, (b) and (d)). The optical axis of the camera intersects the grid plane at its center.

calibration is a one-off procedure that can be used for the rest of the lifetime of the camera/lens combination, provided the optics are not altered.

iv. Once the calibration has been performed, the calibration grid can be removed or covered (if this is necessary to prevent its potential influence on the behaviour of the birds in the experiments).

v. When a bird glides with its wings outstretched, its height (and therefore the 3D coordinates of the wingtips, the thorax point and the head) can be reconstructed in every frame without requiring any interpolation.

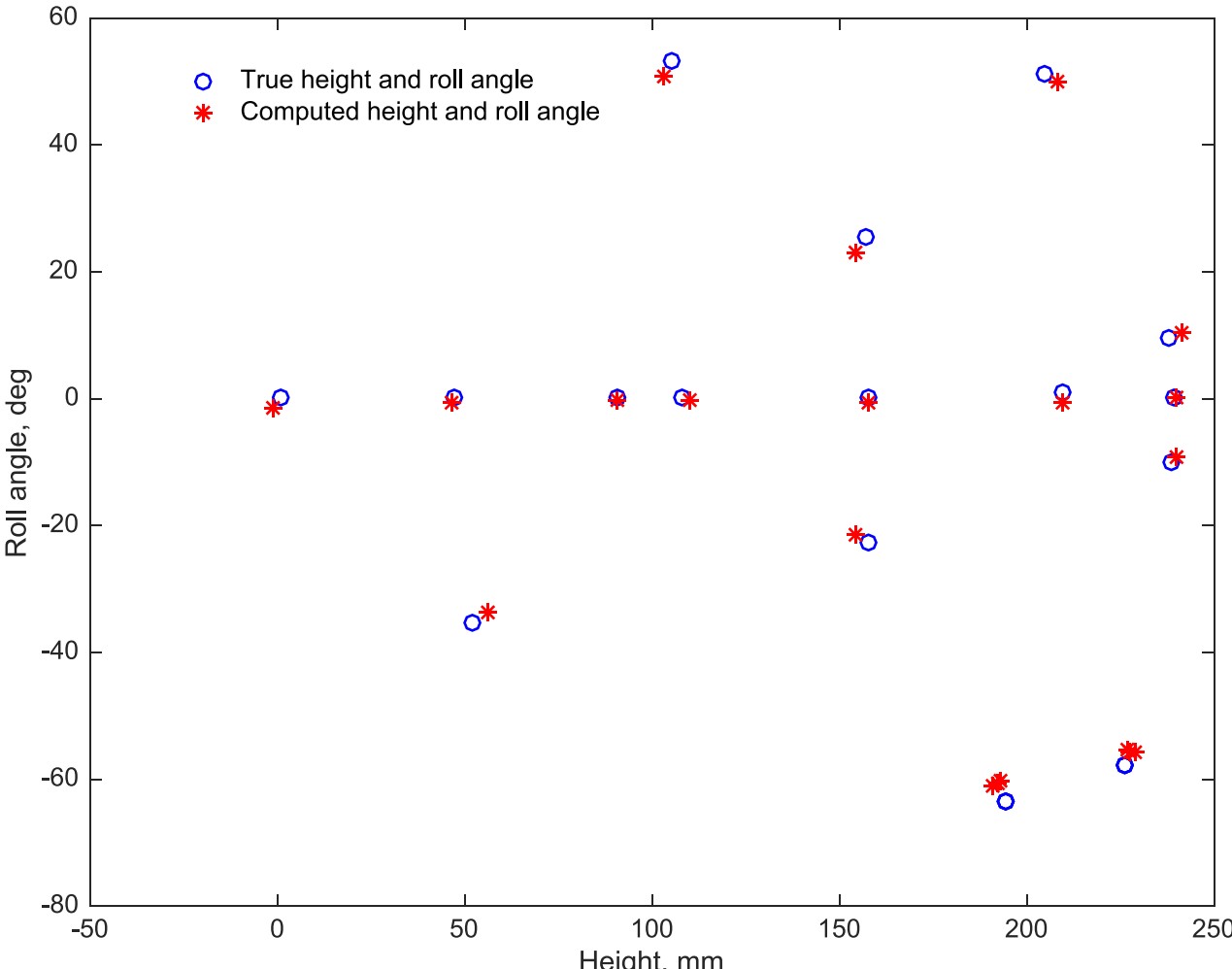

**Fig 13. Comparison of true thorax heights and roll angles with their computed values.** In these test examples the model bird was positioned at 20 different 3D locations within the arena, facing various directions, and at different heights and roll angles.

vi. Moving the camera to a different location does not require recalibration. 3D trajectories of birds can continue to be reconstructed with reference to the new optical axis of the camera and the new plane of the (internally stored) calibration grid. Thus, in principle, the camera that was calibrated in our experiments using the calibration grid on the floor, can also be used to reconstruct the trajectories of birds in outdoor flight by facing the camera upwards and performing the reconstruction relative to the same calibration grid. Trajectory reconstruction is possible even if a bird is located on the opposite side of the calibration grid–the geometry and interpolation underlying the reconstruction will be the same. This is a major attribute of the system, because—unlike systems that use stereo or multiple cameras–it does not need to be recalibrated every time it is moved to a new location.

vii. Because the method is computationally simple, it can be applied in closed-loop experimental paradigms in which visual stimuli need to be modified in real time in response to the bird's flight, as is now being done with some animals (e.g. [26]).

viii. The system is compact, portable, and easily deployed in the field.

ix. It is worth noting that the general method (described in Section C of the S1 File) will work even when the camera's optical axis is not perpendicular to gravity. Regardless of the orientation of the optical axis, the method will deliver the bird's 3D position and roll orientation relative to the plane of the calibration grid, which is perpendicular to the camera's optical axis. The bird's 3D position and roll angle, computed relative to this plane, can then be transformed to the horizontal plane using 3D transformational geometry, from the known orientation of the camera's optical axis relative to the horizontal plane.

The limitations of the method are:

i. We have assumed that the wings are fully extended at each extension, and that the tip-to-tip distance at these extensions is always equal to the measured wingspan. Variability in the wingtip distances from extension to extension (which may occur during certain manoeuvres) will introduce errors in the reconstruction of the 3D trajectory.

ii. The calibration grid on the floor grid must cover a sufficiently large area to enable projection of the wingtips on to the floor at all possible bird positions. This could be a problem when the bird is flying close to the ceiling or to one of the walls of the tunnel (or chamber), as it would require extrapolation of the grid beyond the floor of the chamber. Grid extrapolation can be carried out, but it requires assumptions to be made about the unknown optical distortions in the extrapolated regions of the grid. The calibration grid does not need to be on the floor. It can be in a parallel plane (perpendicular to the camera's optical axis) that is much closer to the camera than the bird–for example, a few centimetres away, to facilitate the calibration of large visual fields. This calibration grid can be superimposed on the bird video images when the thorax point and wingtip positions are being digitized. A camera with a very wide field of view (e.g. from a fish-eye lens) could be calibrated by creating a spherical calibration grid (e.g. lines of latitude and longitude) on a transparent hemispherical shell affixed to the front of the camera, with its centre located at the nodal point of the camera's lens. Alternatively, one could use algorithms that are available for calibrating optical distortions in pinhole cameras (e.g. [27]), provided they are able to capture and characterise all of the relevant optical distortions accurately.

iii. The method requires selection of the *Wex* frames in the video sequence, determination of the pixel co-ordinates of the left and right wingtips and the thorax point in each of the *Wex* frames, and determination of the pixel co-ordinates of the head in each frame of the video sequence. While we have carried out all of these operations manually, they are tedious and time-consuming. Automated tracking and digitisation of the wingtips, the thorax point and the head in the video sequence can be incorporated as an additional 'front end' to the system, which we are currently exploring.

iv. The technique delivers true altitude measurements only at each full wing extension. Altitudes at the intermediate frames are obtained by linear (or spline-based) interpolation. These interpolated heights can be combined with the digitized image position of the head in each frame to obtain a continuous, frame-by-frame 3D trajectory of the bird's head. It is important to note that the X and Y positions of the bird's head are tracked and reconstructed by using new information from *every* frame. The height interpolation should produce reasonably accurate results, provided that the bird's altitude varies smoothly between successive wing extensions. This is very likely to be the case in cruising flight, but may not apply during flight through densely cluttered environments which may entail abrupt changes of altitude as well as variations in the wing kinematics.

v. The roll angle (as we have defined it) is the inclination of the line connecting the wingtips at the time of maximum wing extension. The roll angle is an incidental by-product of our analysis and may not accurately represent the roll angle of the thorax, which could be different. However, this roll angle computation is merely an intermediate step in the reconstruction of the bird's 3D flight trajectory, and does not affect the accuracy of the 3D reconstruction.

vi. The accuracy of the results would deteriorate for birds with small wingspans or birds at a large distance from the camera, because it relies on a precise measurement of the distances between the thorax point and each extended wingtip in the camera image. This issue can be ameliorated to some extent by using a camera with high pixel resolution.

Potential future applications of the method presented in this paper include:

i. Tracking of birds in natural outdoor environments by using an upward-facing wide-angle camera, as discussed briefly above. The species of the bird would have to be known, in order to use an estimate of its wingspan. However, even if the wingspan is not known, roll angles can continue to be computed, and the 3D trajectories of the head and thorax can be reconstructed in units of wingspan. This is demonstrated in Section D of the S1 File. Thus, even if the wingspan of the bird is not known, it is possible to obtain several scale-invariant properties of the bird's trajectory such as its shape, tortuosity, slope of ascent/descent and roll angle, as well as the timing and features of salient temporal events such as the onset of accelerations or decelerations, the frequency of oscillatory movements, or the duration of intermittent gliding phases.

ii. The method can also be applied to reconstruction and analysis of the flight trajectories of multiple birds (for example, in a flock). Again, even if the wingspan is not known, the spatial and temporal properties of the flock can be characterised by specifying inter-bird separations in wingspan units. Difficulties can arise with reconstruction of the 3D tracks of the birds in the flock if their individual wingspans differ significantly, even if they are all of the same species. If the heights of all the birds are approximately the same over a short period of time (or vary only by a small percentage), then one way to deal with this wingspan variability would be to (i) measure the image wingspans ($w_i$) (image distances between the extended wingtips) for each bird over this time period; (ii) compute the mean image wingspan ($w_{mean}$) across all the individuals in the image; (iii) select the individual with the image wingspan ($w^*$) that is closest to the mean image wingspan to be the bird whose physical wingspan matches the known average physical wing span $W_{mean}$ for the species; (iv) estimate the physical wing span $W_i$ of each bird through appropriate scaling based on its image wingspan: $W_i = W_{mean} \cdot (w_i / w^*)$. Once the physical wingspan of each bird has been estimated in this way, the usual technique can be applied to reconstruct the 3D trajectory of each individual in the flock. As mentioned above, this procedure relies on the assumption that the heights of the birds are approximately the same (or vary by only a small percentage) over the short period of time during which their image wingspans are measured. This assumption may not be valid for a flock that flies at a low altitude (because the percentage variation in altitude across individuals could be large), but it is likely to be quite accurate for flocks flying at high altitudes. This is because the percentage variation in altitude across the individuals would then be very small compared to the percentage variation in their wingspans, so that variations in the image wingspans across the birds would then be almost entirely due to variations in their physical wingspans, rather than to variations in altitude. This could be one situation where the (unknown) variability in wingspans across individuals of a bird species can be circumvented by taking advantage of a large sample size.

iii. Reconstruction of 3D flight trajectories of airplanes. In such an application, the 3D coordinates of the airplane and its roll angle can be estimated in every frame without any need for interpolation, because the wingspan is constant (as in a gliding bird). Again, the model of the aircraft would need to be known or identified, in order to use an estimate of its wingspan; otherwise, the aircraft's trajectory can be specified in wingspan units.

## Supporting information

**S1 Checklist. The ARRIVE guidelines 2.0: Author checklist.**
(PDF)

**S1 File.**
(DOCX)

**S1 Video.**
(MP4)

**S2 Video.**
(MP4)

## Author Contributions

**Conceptualization:** M. V. Srinivasan.

**Data curation:** M. V. Srinivasan, H. D. Vo.

**Formal analysis:** M. V. Srinivasan, I. Schiffner.

**Funding acquisition:** M. V. Srinivasan.

**Investigation:** H. D. Vo, I. Schiffner.

**Methodology:** M. V. Srinivasan, H. D. Vo, I. Schiffner.

**Project administration:** M. V. Srinivasan, I. Schiffner.

**Resources:** M. V. Srinivasan.

**Software:** M. V. Srinivasan, H. D. Vo, I. Schiffner.

**Supervision:** M. V. Srinivasan, I. Schiffner.

**Validation:** M. V. Srinivasan.

**Writing – original draft:** M. V. Srinivasan.

**Writing – review & editing:** H. D. Vo, I. Schiffner.

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
