## [Decision Letter · Decision Letter 0]

12 Jun 2022

PONE-D-22-070863D RECONSTRUCTION OF BIRD FLIGHT TRAJECTORIES USING A SINGLE VIDEO CAMERAPLOS ONE

Dear Dr. Srinivasan,

Thank you for submitting your manuscript to PLOS ONE. After careful consideration, we feel that it has merit but does not fully meet PLOS ONE’s publication criteria as it currently stands. Therefore, we invite you to submit a revised version of the manuscript that addresses the points raised during the review process. I appologise for the delay; it took a while to source a world class reviewer with expertise on these topics. I have also carefull read the manuscript and agree with reviewer 1. As this reviewer is so positive about the quality of the research, only a minor revision is required. In doing this please consider if reviewer 1 advice might improve the communication of the research finding. If you can address these points I will be able to proceed promptly with making a final decision.

We look forward to receiving your revised manuscript.

Kind regards,

Adrian G Dyer, Ph.D.

Academic Editor

PLOS ONE

Journal Requirements:

2. As part of your revision, please complete and submit a copy of the Full ARRIVE 2.0 Guidelines checklist, a document that aims to improve experimental reporting and reproducibility of animal studies for purposes of post-publication data analysis and reproducibility: https://arriveguidelines.org/sites/arrive/files/Author%20Checklist%20-%20Full.pdf (PDF). Please include your completed checklist as a Supporting Information file. Note that if your paper is accepted for publication, this checklist will be published as part of your article

Reviewers' comments:

Reviewer's Responses to Questions

**Comments to the Author**

1. Is the manuscript technically sound, and do the data support the conclusions?

Reviewer #1: Yes

2. Has the statistical analysis been performed appropriately and rigorously? 

Reviewer #1: N/A

3. Have the authors made all data underlying the findings in their manuscript fully available?

Reviewer #1: Yes

4. Is the manuscript presented in an intelligible fashion and written in standard English?

Reviewer #1: Yes

5. Review Comments to the Author

Reviewer #1: In their manuscript the authors propose a novel methodology for recovering the position of a bird in a 3D space using data from a single camera. I found the manuscript and method sound and well supported by the geometrical and trigonometrical analyses and formulation. The level of detail is good enough as to allow for replication of the results whilst providing enough support to the author's calculations.

My only minor comments to the manuscript are:

1. More details on the spacing and resolution of the calibrating grid may be of help to others attempting to replicate the method (lines 73-74). Whilst more information about the grid used by the authors to calibrate their system is given later in the manuscript (lines 256-257), readers can benefit from a short discussion on any potential effect of using grids of different dimensions to those presented by the authors. More precisely, is there any potential effect on the calibration if using a grid of dimensions different to those used by the authors? Is there a minimum grid size required to attain an accurate calibration?

2. I think that labels for the y-axis of Figure 8 needs some editing. As the important information here is the position of the thorax and head of the bird (relative to the ground?) for the different slits tested, perhaps the y-axis may use a different label and units.

6. PLOS authors have the option to publish the peer review history of their article (what does this mean?). If published, this will include your full peer review and any attached files.

Reviewer #1: No

---

## [Author Response · Author response to Decision Letter 0]

1 Jul 2022

Dear Editor,

We are glad to hear that the manuscript is potentially acceptable for publication, subject to satisfactory minor revisions. Our responses to the reviewer’s suggestions are outlined below.

1. More details on the spacing and resolution of the calibrating grid may be of help to others attempting to replicate the method (lines 73-74). Whilst more information about the grid used by the authors to calibrate their system is given later in the manuscript (lines 256-257), readers can benefit from a short discussion on any potential effect of using grids of different dimensions to those presented by the authors. More precisely, is there any potential effect on the calibration if using a grid of dimensions different to those used by the authors? Is there a minimum grid size required to attain an accurate calibration?

Thank you for raising this point. While there are benefits to increasing the grid density, there are also some drawbacks. We now discuss the tradeoffs briefly in lines 118-126.

2. I think that labels for the y-axis of Figure 8 needs some editing. As the important information here is the position of the thorax and head of the bird (relative to the ground?) for the different slits tested, perhaps the y-axis may use a different label and units.

We apologize for the confusion that this labeling may have caused. The term “Tunnel width” does not refer to the width of the tunnel, but, rather, to the position (in meters) along the width of the tunnel (corresponding to the y axis in Fig. 3). This notation is used consistently in figures (5-8), and (S1-S6). We have clarified the meaning of the notation by adding an explanation in the legend of Fig. 5, where it is first used (lines 296-298).

Please note that the line numbers cited here for the revisions refer to the text in the revised manuscript per se. They may differ slightly from the line numbers in the version that is compiled by the submission website.

Revisions have also been made in the SI, which now includes a new Section (Section A) that provides the additional information required by the ARRIVE 2.0 Guidelines checklist.

On behalf of all the authors,

Mandyam Srinivasan

Corresponding author

---

## [Editor Report · Decision Letter 1]

5 Jul 2022

3D RECONSTRUCTION OF BIRD FLIGHT TRAJECTORIES USING A SINGLE VIDEO CAMERA

PONE-D-22-07086R1

Dear Dr. Srinivasan,

We’re pleased to inform you that your manuscript has been judged scientifically suitable for publication and will be formally accepted for publication once it meets all outstanding technical requirements.

Kind regards,

Adrian G Dyer, Ph.D.

Academic Editor

PLOS ONE
---

## [Editor Report · Acceptance letter]

25 Jul 2022

PONE-D-22-07086R1 

3D RECONSTRUCTION OF BIRD FLIGHT TRAJECTORIES USING A SINGLE VIDEO CAMERA 

Dear Dr. Srinivasan:

I'm pleased to inform you that your manuscript has been deemed suitable for publication in PLOS ONE. Congratulations! Your manuscript is now with our production department. 

Kind regards, 

on behalf of

Dr. Adrian G Dyer 

Academic Editor

PLOS ONE